# Resolving Cross-Sensitivity Effect in Fluorescence Quenching for Simultaneously Sensing Oxygen and Ammonia Concentrations by an Optical Dual Gas Sensor

**DOI:** 10.3390/s21206940

**Published:** 2021-10-19

**Authors:** Chih-Yi Liu, Moumita Deb, Annada Sankar Sadhu, Riya Karmakar, Ping-Tsung Huang, Yi-Nan Lin, Cheng-Shane Chu, Bhola Nath Pal, Shih-Hsin Chang, Sajal Biring

**Affiliations:** 1Institute of Atomic and Molecular Sciences, Academia Sinica, Taipei 10617, Taiwan; chihyiliu@gate.sinica.edu.tw; 2Department of Electronic Engineering, Ming Chi University of Technology, New Taipei City 24301, Taiwan; moumitadeb644@gmail.com (M.D.); annadamcut@gmail.com (A.S.S.); karmakarriya345@gmail.com (R.K.); jnlin@mail.mcut.edu.tw (Y.-N.L.); biring@mail.mcut.edu.tw (S.B.); 3Organic Electronics Research Center, Ming Chi University of Technology, New Taipei City 24301, Taiwan; 4Department of Chemistry, Fu Jen Catholic University, New Taipei City 24205, Taiwan; 073802@mail.fju.edu.tw; 5Department of Mechanical Engineering, Ming Chi University of Technology, New Taipei City 24301, Taiwan; 6School of Material Science and Technology, Indian Institute of Technology (BHU), Varanasi 221005, India; bnpal.mst@itbhu.ac.in; 7MSSCORPS CO., LTD., Hsinchu 30072, Taiwan; Shih-Hsin.Chang@msscorps.com

**Keywords:** dual gas sensor, optical gas sensor, cross-sensitivity, fluorescence-based sensor, fluorescence quenching, PtTFPP, eosin Y

## Abstract

Simultaneous sensing of multiple gases by a single fluorescent-based gas sensor is of utmost importance for practical applications. Such sensing is strongly hindered by cross-sensitivity effects. In this study, we propose a novel analysis method to ameliorate such hindrance. The trial sensor used here was fabricated by coating platinum(II) meso-tetrakis(pentafluorophenyl)porphyrin (PtTFPP) and eosin-Y dye molecules on both sides of a filter paper for sensing O_2_ and NH_3_ gases simultaneously. The fluorescent peak intensities of the dyes can be quenched by the analytes and this phenomenon is used to identify the gas concentrations. Ideally, each dye is only sensitive to one gas species. However, the fluorescent peak related to O_2_ sensing is also quenched by NH_3_ and vice versa. Such cross-sensitivity strongly hinders gas concentration detection. Therefore, we have studied this cross-sensitivity effect systematically and thus proposed a new analysis method for accurate estimation of gas concentration. Comparing with a traditional method (neglecting cross-sensitivity), this analysis improves O_2_-detection error from −11.4% ± 34.3% to 2.0% ± 10.2% in a mixed background of NH_3_ and N_2_.

## 1. Introduction

Many technologies have been developed for gas sensing, which sees wide applications in various fields such as environmental contaminant detection [1,2,3,4,5,6]. One of them uses the change in fluorescence intensity of dye molecules in presence of the target analyte gas molecules. Such sensing has attracted great attention because it provides multiple information by detecting optical parameters such as intensity, polarization, decay time, energy transfer, and quenching efficiency [7,8,9,10,11]. In addition, fluorescence-based gas sensing is more popular compared to other spectroscopic methods based on optical measurements of absorption, reflection, interference, Raman scattering, and surface plasmon resonance [9]. Nowadays, numerous fluorescence-based gas sensors have been developed for potential applications [12,13,14,15,16].

With the continual improvement demands from the industry, it is desirable to have a sensor capable of detecting two or more different gases simultaneously. The sensor must possess the ability to identify not only species but also the concentration of sensed gases. Such desirability can be fulfilled by a fluorescence-based gas sensor fabricated with several different dyes sensitive to individual analyte gas species [11]. Ideally, each dye produces one or more fluorescence peaks whose intensity can be quenched/enhanced in the presence of a specific gas species. Therefore, the peak intensity variation can be used to determine whether a specific gas is present or not. Furthermore, the variation level can be applied to identify the concentration of the gas species. However, such peak specificity may be incorrect in a real situation, where a fluorescence peak can be quenched/enhanced by two or more different gases simultaneously [17]. Such cross-sensitivity effects strongly hinder the gas detection task, particularly with respect to the accuracy of gas concentration identification. The more gas species are sensed, the more complex the cross-sensitivity created is, leading to severe detection hindrance. Therefore, resolving cross-sensitivity effects is crucial to the development of a fluorescence-based gas sensor with multi-analyte detection ability. Here, we present a systematic study on the cross-sensitivity effect of a fluorescence-based dual gas sensor which detects oxygen and ammonia simultaneously. According to the systematic study, we propose an analysis method to strongly improve the gas concentration detection accuracy in presence of cross-sensitivity effects.

Oxygen is a colorless and odorless gas and is essential to the environment, oceans, agriculture, industry and health. An oxygen concentration range of 19.5–23.5% in the environment is vital for living life [18]. On the other hand, ammonia also plays a crucial role in agriculture, bioprocessing and food-freshness testing. Its vapor hurts the eyes (>50 ppm) and respiratory system (>500 ppm) of humans [19]. Therefore, many researchers have focused on the development of oxygen and ammonia sensors [20,21,22,23,24,25,26,27,28]. Recently, we have reported a fluorescence-based dual gas sensor with detection sensitivities of 60 for oxygen and 20 for ammonia [29]. However, this sensor suffers from cross-sensitivity effects and thus fails to properly detect the concentration of the individual gases. Such a drawback could probably be overcome by using the analysis method presented here. Furthermore, it is promising to apply this analysis method for improving the accuracy in the detection of concentration of various fluorescence-based multi-gas sensors.

## 2. Experimental

### 2.1. Chemical Materials

The chemicals used in this study are as follows: Grade 1 filter paper was obtained from Advantec (Tokyo, Japan), platinum(II) meso-tetrakis(pentafluorophenyl)porphyrin (PtTFPP) from Frontier Scientific (Logan, UT, USA). Triton-X100 (analytical grade, 100%) and tetraethylorthosilane (TEOS, 99.5%) were from Acros Organics (Geel, Belgium), *n*-octyltriethoxysilane (Octyl-triEOS, 97.5%) was from Alfa Aesar (Haverhill, MA, USA) and cellulose acetate (CA) powder from Showa Chemicals (Akasaka Minato-Ku, Japan). Other reagents such as EtOH (99.5%), SiO_2_ (99.9%) were purchased from ECHO Chemical Co., Ltd. (Miaoli, Taiwan) and tetrahydrofuran (THF, 99.9%) was from TEDIA (Fairfield, OH, USA). Eosin-Y (99%) and acetic acid (99%) were purchased from Sigma Aldrich (St. Louis, MO, USA) and HCl (32%) from Shimakyu (Taichung, Taiwan). All the chemicals were used as received without further purification.

### 2.2. Trial Sensor Fabrication

The flowchart in Figure 1a schematically shows the procedures to synthesize oxygen- and ammonia- sensing solutions. 0.05 g of PtTFPP (oxygen-sensing material [30]) was dissolved in 10 mL of THF to form a dye solution. Thirty μL of this solution was mixed with 30 μL of a sol-gel matrix. This mixture was stirred magnetically for 10 min to create an oxygen-sensing solution. The matrix was prepared as follows: we mixed 4 mL of TEOS and 0.4 mL of Octyl-triEOS to form a precursor solution. After that, EtOH (1.25 mL) and then HCl (0.4 mL) were added to the solution. The mixture was capped and stirred magnetically at room temperature for 1 h, during which Triton-X-100 (0.2 mL) was added to the mixture. Finally, the sol-gel matrix was created. 0.05 g of eosin-Y (the ammonia sensing material [31]) was dissolved in 10 mL of THF to form a dye solution. One hundred μL of the dye solution was mixed with 50 μL of another matrix mixture and 2 mg of SiO_2_ nanoparticles. This mixture was stirred magnetically for 10 min to create an ammonia-sensing solution. The matrix was created by dissolving 0.22 g of CA powder in 10 mL acetic acid and stirred magnetically at 40 °C for 1.5–2 h to form a transparent solution.

With their large surface areas, porous materials have the benefit of adsorbing numerous detected species and thus are widely applied for various sensing tasks [32,33,34,35,36,37]. Filter paper is one of the most commonly used porous materials. Here, this material was used to carry the sensing materials for a trial sensor. The fabrication concept of the dual sensor of gases (O_2_ and NH_3_) is schematically represented in Figure 1b. We dropped 100 μL of ammonia-sensing solution on one side (bottom side) of a piece of filter paper (thickness of 200 μm). The sample was then dried in air at room temperature. After that, a similar process was used again to treat the other side (topside) of the paper with 30 μL of the oxygen-sensing solution. Then this sample was dried at room temperature for 24 h to obtain effective sensing materials. With the sensing materials, the sample functioned as a fluorescent-based dual sensor for simultaneously detecting gases of O_2_ and NH_3_.

The filter paper used here was made of many entwined fibers which formed a porous structure, as shown in the typically topside SEM image of Figure 2a. Such a structure contains huge exposed surfaces allowing it to to absorb other materials. After treating with sensing solutions, the sample’s fiber surfaces were fully covered by sensing materials, as shown by the typically topside SEM image of Figure 2b. In fact, the sensing solution penetrated the paper in the sensor fabrication process. Therefore, increasing the sensor thickness allows it to absorb more sensing material, leading to stronger fluorescence signals in the subsequent gas detection process.

### 2.3. Optical Sensing Instruments

The instrumental setup for optical sensing is schematically depicted in Figure 3. The sample was excited by a UV light-emitting diode (LED) with a central wavelength of 405 nm driven by a generator with an arbitrary waveform (TGA1240, Thurlby Thandar Instruments (TTI) Ltd., Huntington, UK) at 10 kHz. A fiber optics spectrometer of USB4000 (Ocean Optics Inc., Largo, FL, USA) was employed to measure emission spectra from the trial sensor in the sample chamber.

O_2_, NH_3_, and N_2_ flowed into and then out of the sample chamber continuously to control its atmosphere and thus modified emission spectra. Prior to the flow, the gases were ejected into a mixing chamber through mass flow controllers (Model GFC 17, Aalborg Instruments and Controls Inc., Orangeburg, NY, USA) at room temperature. The controllers were able to precisely adjust the flow rate of each gas species and thus set the environmental concentrations of the three gases. In this paper, only the concentrations of O_2_ and NH_3_ are indicated since that of the residual N_2_ can be calculated easily. In fact, N_2_ was too inert to react with sensing materials and thus barely affected the experimental results. The concentration unit used for O_2_ is percentage (%) while that for NH_3_ is ppm. This is because the two units are commonly used for corresponding sensors from a practical point of view to decide whether the environment is harmful to health.

## 3. Results and Discussion

### 3.1. Analysis Theorem

The sensing materials used in this study are fluorophore compounds which contains a number of active sites to emit fluorescence upon light excitation. Depending on compound characteristics, active sites can be occupied by specific gas molecules. Such occupied sites that quench the fluorescence of compounds and related gas molecules are called quenchers. The concentration of quenchers dominates the degree of quenching. Therefore, detection sensitivity, employed to judge the quality of a sensor, can be determined as I_0_/I, where I_0_ and I represent the steady-state fluorescence intensities in the absence and presence of quencher. Previous studies indicate that the detection sensitivity follows the Stern-Volmer (S-V) equation [38]:I_0_/I = 1 + K_SV_[Q](1)
where K_sv_ and [Q] are the Stern-Volmer quenching constant and quencher concentration, respectively. This equation considers the change in fluorescence for an ideal case where all the sensing molecules are sensitive to the quencher, however, in a real situation a number of molecules may be insensitive. Therefore, the equation is rewritten as the modified S-V equation [38,39,40]:I_0_/I = [{f/(1 + K_SV_[Q])} + (1 − f)]^−1^(2)
where K_sv_ and [Q] are defined as those in Equation (1); f denotes the fraction of the fluorescence caused by the sensitive molecules in a quencher-free environment. The modified S-V equation properly fitted our results and thus was used to analyze the experimental data.

### 3.2. Emission Spectra under Different O_2_ and NH_3_ Concentrations

Figure 4a shows the emission spectra from the trial dual sensor obtained by systematically varying the O_2_ concentration in a NH_3_-free environment. The fluorescence peak of PtTFPP dye at 650 nm (called “O_2_-sensitive peak” hereafter) is used to detect O_2_ since its intensity clearly reduces upon increasing the gas concentration. Similarly, the fluorescence peak of eosin-Y at 580 nm is sensitive to NH_3_ in an O_2_-free environment (called “NH_3_- sensitive peak” hereafter), as revealed in Figure 4b. The spectrum (the inset in Figure 4b) points out that the peak intensity was reduced gradually with the increase in NH_3_ concentration. However, the change in NH_3_ concentration alters not only this peak, but also the O_2_-sensitive one as shown in Figure 4b. In fact, the NH_3_-sensitive peak is also quenched by O_2_ as shown in the inset of Figure 4a. The combined observation from Figure 4a,b and their insets imply that both the O_2_ and NH_3_ peaks suffer from cross-sensitivity when the environmental atmosphere simultaneously contains O_2_ and NH_3_ gases. For example, Figure 4c and its inset show the emission spectra of the dual sensor by systematically varying O_2_ concentration in an environment containing 200 ppm of NH_3_. Compared with the ammonia-free case (Figure 4a), both the O_2_ and NH_3_-sensitive peaks show less intensities for a given oxygen concentration indicating that the cross-sensitivity effect needs to be addressed in gas sensing.

Cross-sensitivity effects strongly hinder the quantitative detection of gas concentrations since more than one gas species may contribute to the intensity variation of a sensing peak. Such quantitative detection is indispensable for a gas sensor for practical applications and thus developing an analysis method for the quantification is highly demanded. Such analysis is based on highly accurate data, however, the O_2_- and NH_3_-sensitive peaks are too close to be well-separated (Figure 4a,c), thus hindering the collection of accurate data. The situation gets even worse since additional peaks, e.g., the one around 710 nm interfere with the intensity measurement. Therefore, extracting accurate peak intensities is necessary for further data analysis and will be presented in the followed subsection. A systematic study on PtTFPP peaks of 650 and 710 nm varying with different oxygen and ammonia concentrations is presented in Appendix A.

### 3.3. Spectra Fitting

We employed Gaussian fitting to analyze the measured emission spectra for extracting more accurate positions and intensities of fluorescent peaks. This analysis is applicable to spectra corresponding to any concentration of O_2_ and NH_3_. For example, Figure 5 shows an emission spectrum (black curve) under conditions of 40% of O_2_ and 400 ppm of NH_3_. A Gaussian fitting indicates that the spectrum can be separated into four different parts which are displayed by the red, light green, blue, and light blue curves in Figure 5, respectively. Addition of the four fitted curves together creates the purple one which is quite similar to the original spectrum. The four curves have four peaks marked as “Peaks 1–4” in Figure 5. Peaks 3 and 4 originate in the material for oxygen sensing while the others come from that used for ammonia detection. Each gas species only needs one peak to monitor its concentration. In other words, only two peaks are required for the current gas mixture sensing. We selected Peak 1 (called “fitted NH_3_-sensitive peak” hereafter) for NH_3_ sensing and Peak 3 (called “fitted O_2_-sensitive peak” hereafter) for O_2_ detection because of their higher peak intensities. The variation in the intensity of the peaks reflects the change in gas concentrations and will be discussed later.

### 3.4. Spectral Analysis for Single Gas Species

Prior to studying the cross-sensitivity effect, we need to understand how sensing peaks change with single gas species. Such study is relatively simple and can provide crucial information for the exploration of complicated cross-sensitivity effects. Figure 4a (O_2_ only) and 4b (NH_3_ only) shows the emission spectra for such a study. Gaussian fitting was used for the spectra in Figure 4b to obtain fitted NH_3_-sensitive peaks for different ammonia concentrations while those in Figure 4a were used to acquire fitted O_2_-sensitive peaks for distinct oxygen concentrations. In addition, the fitted peaks were employed to calculate the detection sensitivity of the corresponding gas species (refer to Section 3.1). Figure 6a shows the plot of the sensitivity as a function of the ammonia concentration in an oxygen-free (NH_3_ only) environment. The increasing trend of the plot quantitatively indicates the ability of fluorescence quenching caused by ammonia. The maximum detection sensitivity is 4.8 for a NH_3_ concentration of 1000 ppm. A similar increasing trend was also observed in the O_2_-only case, as shown in the plot of the sensitivity as a function of the oxygen concentration in an ammonia-free environment (Figure 6b). The maximum detection sensitivity is 47 for an O_2_ concentration of 100%.

Equation (2) was used to fit the measured sensitivity-concentration data, as shown in the red curves in Figure 6a (NH_3_ only) and Figure 6b (O_2_ only). The fittings are correct since their coefficients of determination are both larger than 0.99. The fitting indicates parameters of f = 0.84 and K_SV_ = 0.14 ppm^−1^ for the ammonia-only case and f = 0.99 and K_SV_ = 0.71%^−1^ for the oxygen-only one. For the ammonia-only case, the f of 0.84 deviates somewhat from 1, implying that ~16% of ammonia-sensing dye is insensitive to NH_3_. For the other case, the f of 0.99 quite close to 1 implies that most of oxygen-sensing dye molecules are sensitive to O_2_. The fitted f and K_SV_ used to estimate the gas concentration of the sensed atmosphere will be discussed later.

### 3.5. Systematic Study of Cross-Sensitivity Effect

Mixtures of two gases, i.e., oxygen and ammonia, were used in this study (the influence of nitrogen is considered negligible). To explore the cross-sensitivity effect, we measured emission spectra from the trial sensor under systematically varied concentrations of O_2_ and NH_3_. The spectra were then analyzed by a method similar to that described in Section 3.4 to acquire the corresponding sensitivity, f, and K_SV_ values. Figure 7a shows the plot of sensitivity of the fitted NH_3_-sensitive peak as a function of ammonia concentration under different oxygen environments. The sensitivity varies with different oxygen environments for a fixed ammonia concentration. The relation between sensitivities and ammonia concentrations fits Equation (2), no matter under which oxygen environment, as shown by the colored curves in Figure 7a. Figure 7b shows values of f and K_SV_ as functions of the O_2_ concentration based on the fitting curves in Figure 7a. The parameter f has an average value of 0.79 and standard deviation of 0.07, which implies f fluctuates within ~±10%. In addition, no clear correlation between f and oxygen concentration is observed in Figure 7b (red squares). Therefore, we infer that environmental oxygen gas does not significantly change the amount of sensitive dye molecules for NH_3_ sensing. Unlike f, K_SV_ monotonically decreases with increasing oxygen concentration, as shown by the blue dots in Figure 7b. Thus, the coupling between NH_3_ molecules and ammonia-sensing dyes is reduced by environmental oxygen gas. The higher oxygen concentration leads to a lower coupling. The maximum K_SV_ (0.014 ppm^−1^ at 0% O_2_) is seven times the minimum one (0.002 ppm^−1^ at 80% O_2_), as shown by the blue dots in Figure 7b. Such a large variation of K_SV_ implies that the coupling can be strongly modified by background oxygen. As a result, we speculate that oxygen-induced coupling reduction is the main reason causing the sensitivity changes observed in different oxygen environments for a fixed ammonia concentration, as shown in Figure 7a.

We also systematically studied the sensitivity of the fitted O_2_-sensitive peak as a function of the oxygen concentration under different ammonia environments, as shown in Figure 8a. This figure indicates that the relation between sensitivities and oxygen concentrations fits Equation (2) no matter under which ammonia environment is used, as shown by the colored curves in Figure 8a. We obtained values of f (red squares in Figure 8b) and K_SV_ (blue dots in Figure 8b) as functions of NH_3_ concentrations based on the fitting curves in Figure 8a. The parameter f has an average value of 0.98 and standard deviation of 0.005, which imply that f fluctuates within a small range of ~±0.5%. Such a small standard deviation implies that f barely changes for any NH_3_ environment. In addition, the f value is quite close to one, implying that most of the oxygen-sensing molecules are sensitive to the O_2_ concentration. In other words, an ammonia environment barely changes the amount of oxygen-sensitive dye molecules. Roughly speaking, this inference is similar to that observed for the case of the fitted NH_3_-sensitive peak presented in the last paragraph. Unlike f, a significant change in K_SV_ values with different ammonia concentrations implies that the coupling between oxygen-sensing molecules and O_2_ is modified by the NH_3_ environment. The maximum K_SV_ of 0.71%^−1^ is observed at a NH_3_-free environment. The K_SV_ substantially reduces to 0.3%^−1^ for a NH_3_ concentration of 100 ppm and maintains approximately the same value of 0.5%^−1^ for higher NH_3_ cases, as shown by the blue dots in Figure 8b. Such a trend is qualitatively different with that of the fitted NH_3_-sensitive peak presented in the last paragraph. Following a discussion similar to that for Figure 7a presented in the last paragraph, we again speculate the ammonia-induced coupling variation is responsible for the sensitivity changes observed in different ammonia environments for a fixed oxygen concentration, as shown in Figure 8a. The study of the cross-sensitivity presented in this subsection provides crucial information for sensing concentrations of oxygen and ammonia and will be discussed in the following subsection.

### 3.6. Estimation of Gas Concentration

The main goal of our study was to develop a method to improve gas concentration estimations of sensing methods with cross-sensitivity effects. The process starts by measuring an emission spectrum from a sensed atmosphere to obtain fitted O_2_- and NH_3_-sensitive peaks (refer to Section 3.3). The fitted peaks are then used to calculate the sensitivities. We tried to neglect any cross-sensitivity effect and used the values of f and K_SV_ presented in Section 3.4 to analyze the sensitivities because of the relatively simple process. The f and Ksv values together with the calculated sensitivities were substituted into Equation (2) to estimate the ammonia and oxygen concentrations. This analysis method is called hereafter the direct method. We arbitrarily selected seven cases of different oxygen and ammonia concentrations for testing the accuracy of estimated gas concentrations by the direct method, which resulted in the errors show in Table 1. The error is calculated as (real concentration-estimated concentration)/(real concentration) where the real concentration is controlled by the experimental setting. This table indicates an average error of −1.2% and standard deviation of 4.2% for NH_3_ sensing. In general, a scientific measurement displaying an error within ~±5% is considered acceptable. However, the O_2_ sensing analysis leads to an average error of −11.4% and standard deviation of 34.3%, i.e., the accuracy is too poor to be acceptable. Therefore, the analysis method to estimate O_2_ concentration needs to consider cross-sensitivity effect for better accuracy.

As mentioned above, the direct method is able to provide NH_3_ concentrations with acceptable errors, however, the determination of oxygen concentrations needs to take into account of cross-sensitivity effect, which causes f and K_sv_ for O_2_ sensing to be different from that in a NH_3_-free environment (Figure 8b). Thus, we used the direct method to estimate ammonia concentrations in any environment under study. Then this concentration viewed as the NH_3_ background was employed to determine f and K_sv_ for O_2_ sensing by an interpolation method using the data in Figure 8b. The determined f and K_sv_ together with the calculated sensitivity corresponding to the fitted O_2_-sensitive peak were then substituted into Equation (2) to estimate the accurate oxygen concentration. This analysis method, called modified method hereafter, was used to estimate oxygen concentrations for the test cases (environments with different mixture of O_2_ and NH_3_ gases) in Table 1. The absolute value of the error for the oxygen concentration estimation by this method is dramatically smaller than that obtained by the direct method, as presented in Table 1. Comparing with the direct method, this analysis improves the average error from −11.4% to 2.0% and the standard deviation from 34.3% to 10.2%. Figure 9 shows the plot of estimated oxygen-concentration errors as a function of case number for the direct (blue square) and modified (red dots) methods. This figure clearly indicates that the error for each case obtained by the modified method in comparison with that obtained by the direct one is notably closer to 0. Such a prominent improvement indicates that the modified method is indeed useful and probably can be applied for other fluorescence-based analyte sensing tasks. It is worthwhile to note that all the gaseous mixtures for the testing cases in Table 1 are different from those for the data points in Figure 7 and Figure 8 to guarantee the accuracy of modified method in a completely unknown atmosphere.

Although the modified method reduces the detection error, it may be still insufficient for practical applications. Developing proper dyes for sensors with low cross-sensitivity is needed. The modified method can help such sensors reduce the detection error to achieve a qualified accuracy. Using a single sensor instead of many sensors for multi-gas detection has many advantages such as cost reduction and the possibility of device miniaturization, however, cross-sensitivity effects strongly hinder the development of fluorescence-based multi-gas sensors and thus more work contributions on devices for single species detection is needed. The current analysis sheds some light to help researchers overcome the hindrance. In addition, this analysis is only applied for two-gas detection. Detecting gas mixtures containing more than two species may be required in many situations. The cross-sensitivity effects for such a detection may be more complicated. A systematic study similar to that presented here could help to resolve the complexity and thus provide crucial information for the development of fluorescence-based multi-gas sensors.

## 4. Conclusions

Fluorescence-based gas sensors have many advantages such as high detection sensitivity and cost effectiveness. It is even better if such sensors have the ability to sense multiple gases simultaneously because several species may coexist in many practical applications. A multi-gas sensor needs to identify not only the species but also the concentration of the detected gases. Such a sensor can be fabricated by using several distinct fluorescent dyes, each of which is sensitive to only one specific gas species. However, a real sensor may not have such specificity; the dye used for this sensor may be sensitive to more than one species. Such a phenomenon, called a cross-sensitivity effect, strongly hinders the development of fluorescence-based multi-gas sensors. In this work we systematically studied such an effect by using a trial fluorescence-based sensor which allowed us to sense oxygen and ammonia gases simultaneously. According to this study, we proposed a new analysis method to reduce the cross-sensitivity effect and thus improve the accuracy of gas concentration detection. This method has been tested by sensing seven arbitrarily selected atmospheres with different compositions of ammonia and oxygen gases. This analysis improves oxygen-detection error from −11.4% ± 34.3% to 2.0% ± 10.2% in a mixed background of ammonia and nitrogen when compared with that neglecting the cross-sensitivity effect. Such an analysis method could probably be applied for other fluorescence-based multi-gas sensors to resolve their cross-sensitivity effects. Therefore, the proposed method is promising for the development of multi-gas sensors with higher accuracy in the detection of gas concentrations in real environments.

## Figures and Tables

**Figure 1 sensors-21-06940-f001:**
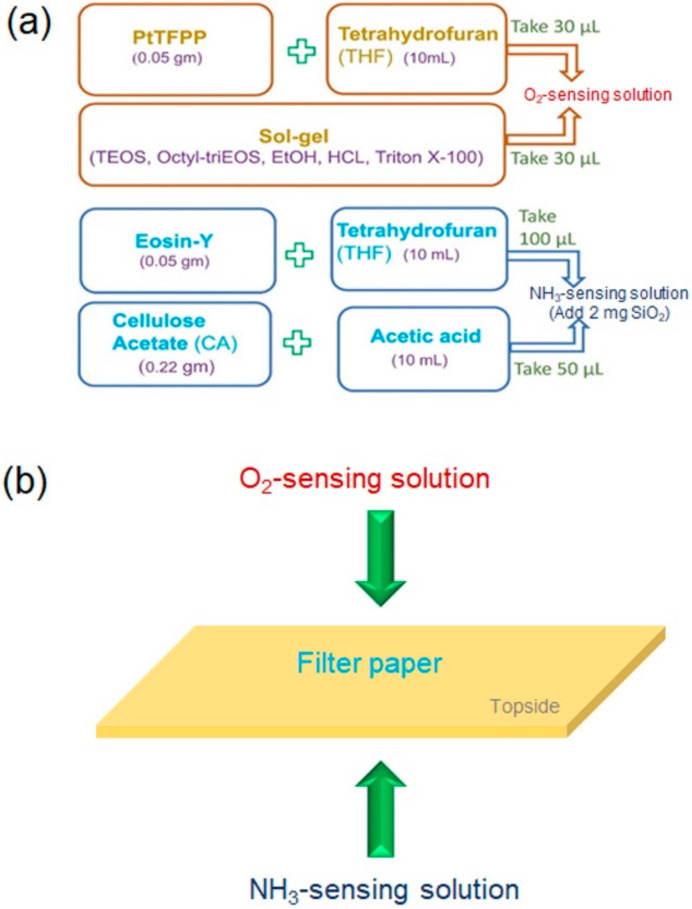
(**a**) A flow chart showing the synthesis processes of O_2_- and NH_3_-sensing solutions. (**b**) Schematic diagram representing the fabrication concept of a trial dual sensor.

**Figure 2 sensors-21-06940-f002:**
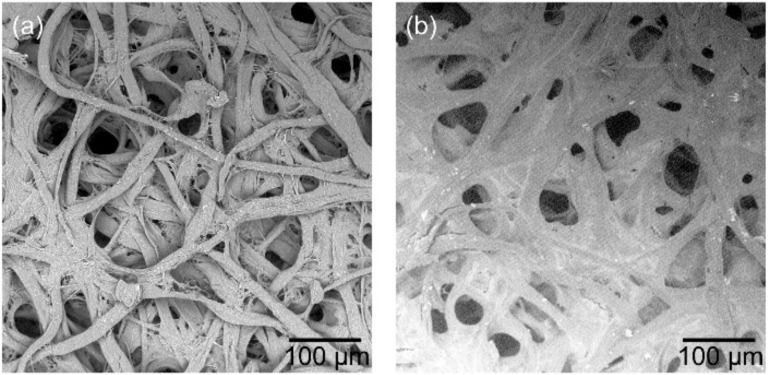
Topside SEM images of a piece of filter (**a**) before and (**b**) after treated with sensing solutions. The treatment process is schematically represented in Figure 1.

**Figure 3 sensors-21-06940-f003:**
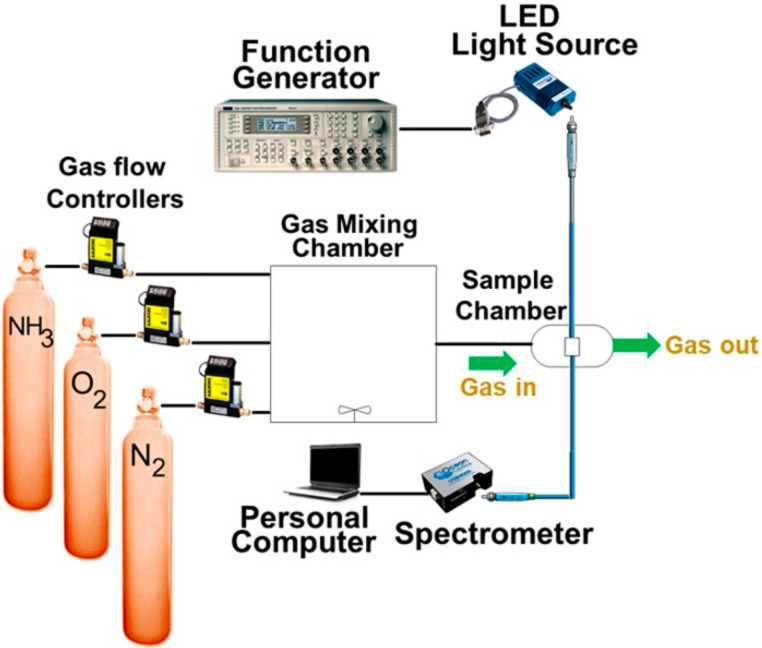
Schematic diagrams of the system setup for optical gas sensing.

**Figure 4 sensors-21-06940-f004:**
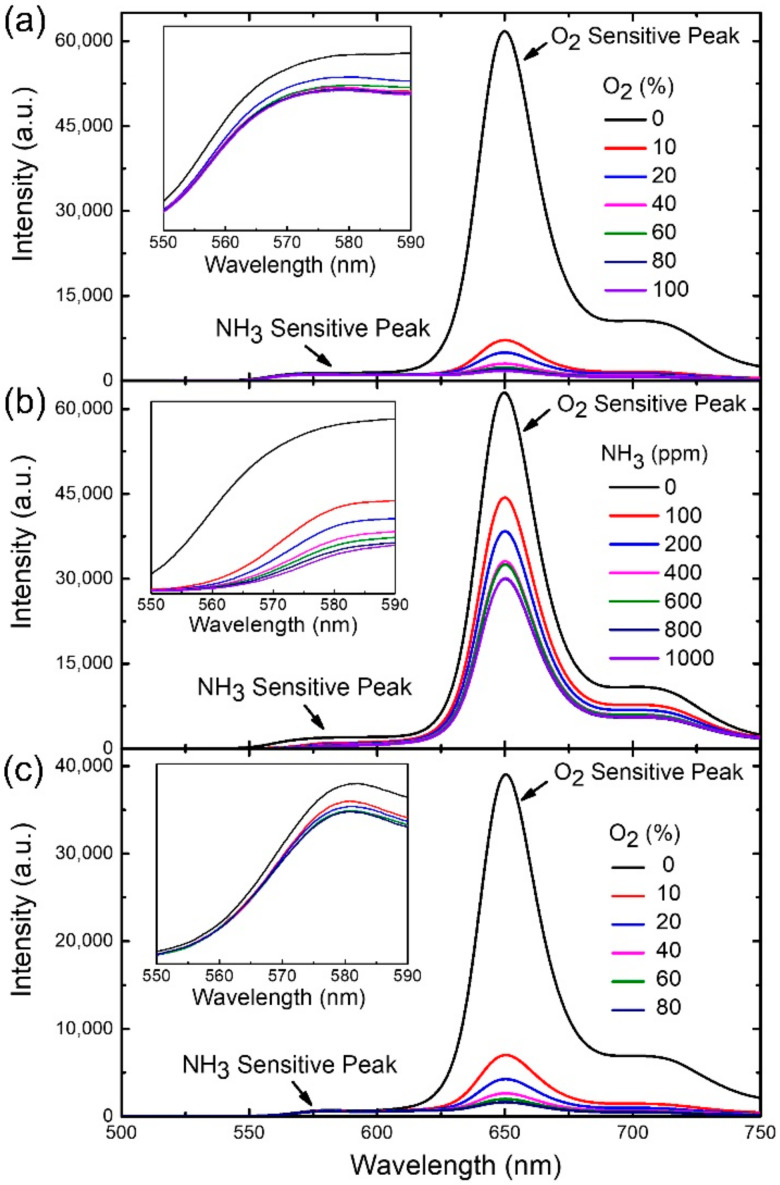
Systematic study of the emission spectra of a trial dual sensor under (**a**) 0 ppm of NH_3_, (**b**) 0% of O_2_ and (**c**) 200 ppm of NH_3_. The insets show the enlarged areas for NH_3_-sensitive peaks in the corresponding spectra. The intensity units for the insets are arbitrary units.

**Figure 5 sensors-21-06940-f005:**
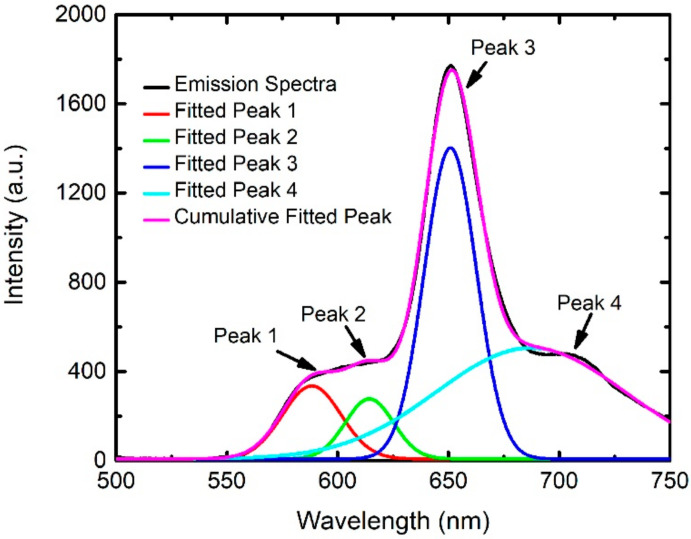
A typical example showing the Gaussian fitting of the emission spectra under conditions of 40% O_2_ and 400 ppm NH_3_.

**Figure 6 sensors-21-06940-f006:**
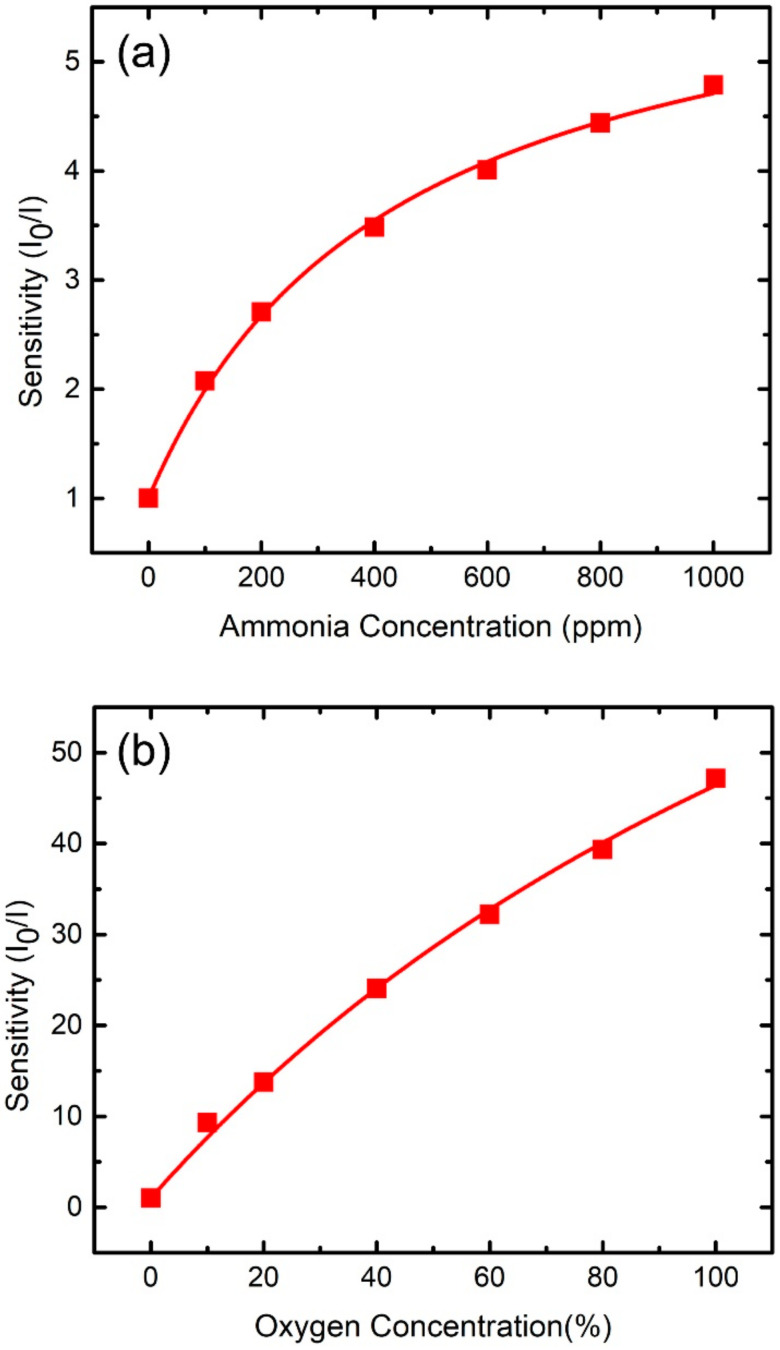
Sensitivity (I_0_/I) of (**a**) fitted NH_3_-sensitive peak as a function of ammonia concentration under an oxygen-free environment and (**b**) fitted O_2_-sensitive peak as a function of oxygen concentration under an ammonia-free environment. Equation (2) is used to fit the data points as shown by the red curves.

**Figure 7 sensors-21-06940-f007:**
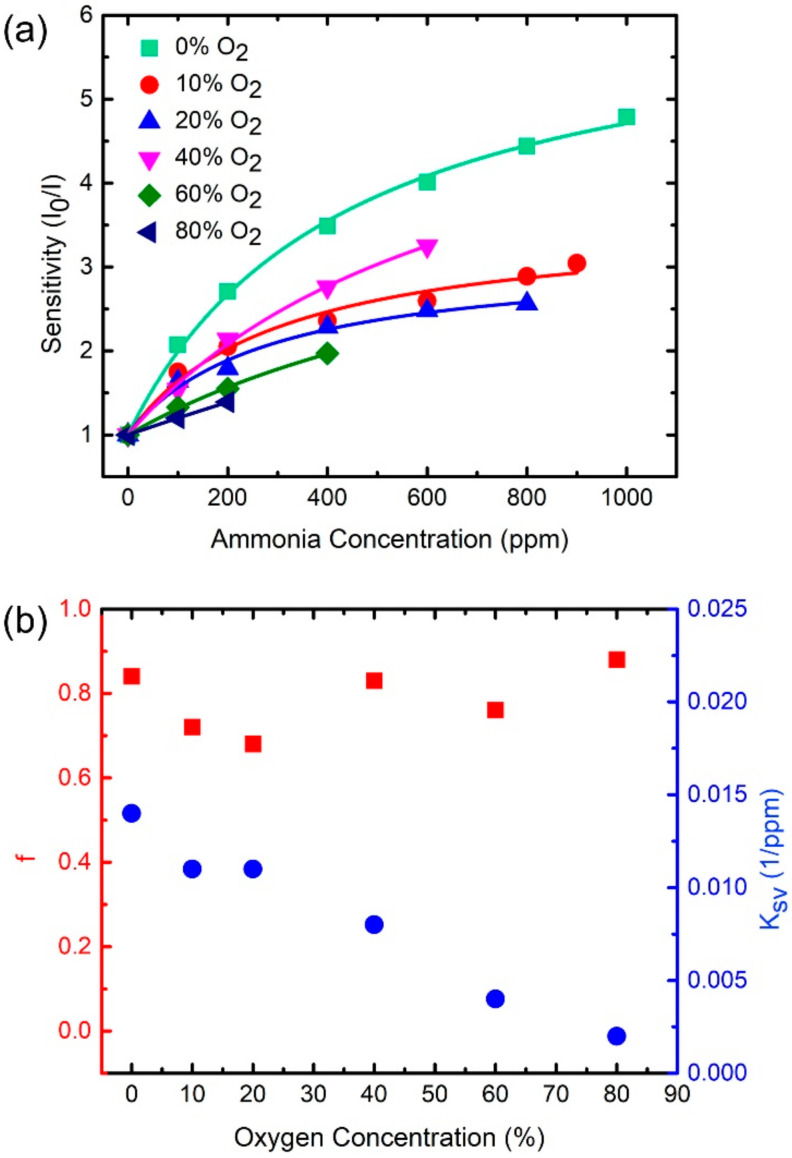
(**a**) Sensitivity (I_0_/I) of fitted NH_3_ sensitive peak as a function of ammonia concentration under systematically varying environmental oxygen concentration. Equation (2) is used to fit the data points as shown by the colored curves. (**b**) f (red squares) and K_SV_ (blue dots) as a function of oxygen concentration based on the fitted colored curves in (**a**). The f and K_SV_ are parameters in Equation (2).

**Figure 8 sensors-21-06940-f008:**
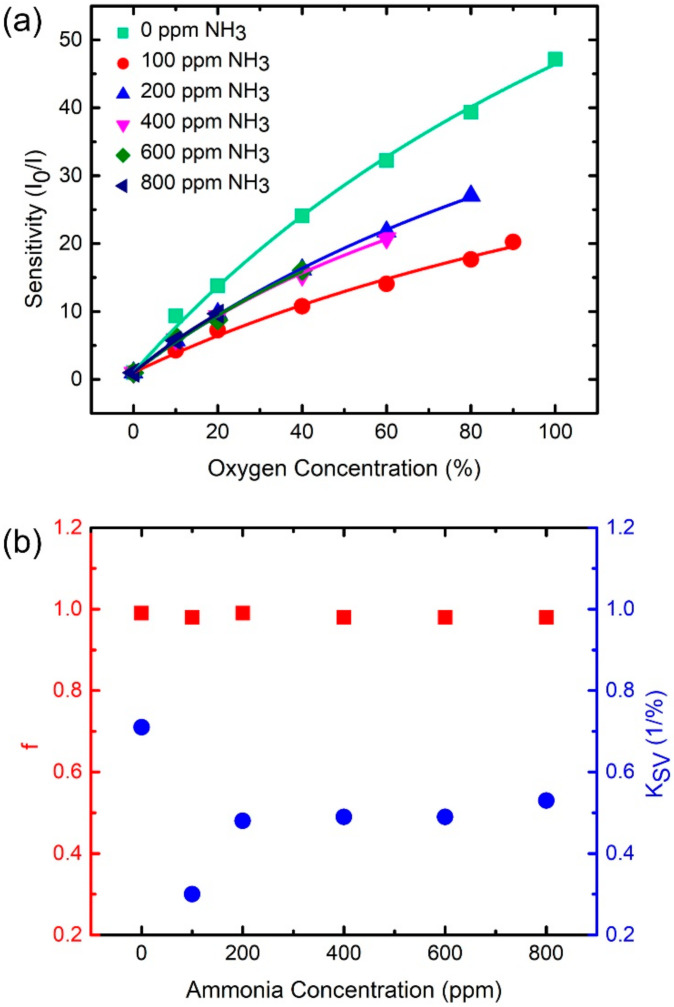
(**a**) Sensitivity (I_0_/I) of a fitted O_2_-sensitive peak as a function of oxygen concentration under systematically varying environmental ammonia concentrations. Equation (2) is used to fit the data points, as shown by the colored curves. (**b**) f (red squares) and K_SV_ (blue dots) as a function of ammonia concentration based on the fitted colored curves in (**a**). The f and K_SV_ are parameters in Equation (2).

**Figure 9 sensors-21-06940-f009:**
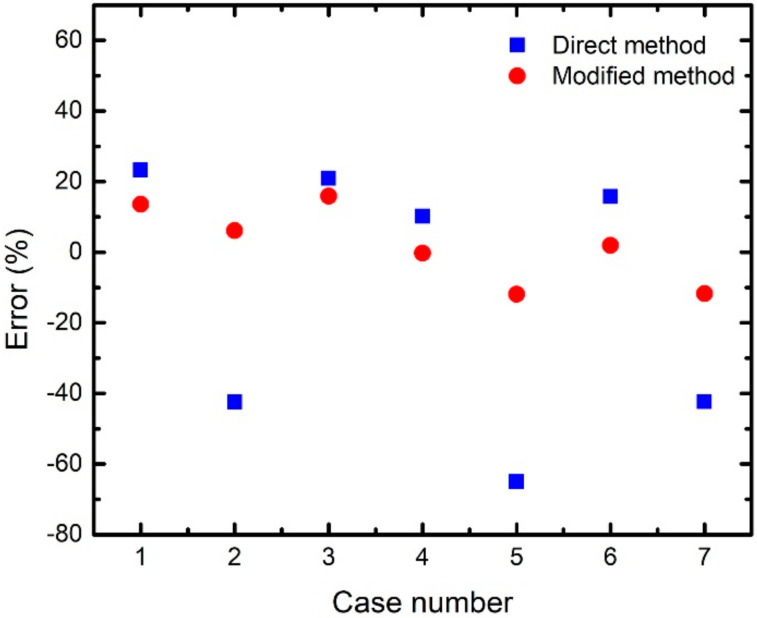
Estimated O_2_-concentration error as a function of case number for the direct (blue squares) and modified (red dots) methods. The experimental conditions of the various cases are presented in Table 1.

**Table 1 sensors-21-06940-t001:** Error of quantitative analysis for gas concentration.

Case Number	1	2	3	4	5	6	7
Real NH_3_ concentration (ppm)	50	500	150	150	700	50	500
Real O_2_ concentration (%)	5	5	10	20	20	30	50
NH_3_-concentration error by the direct method (%)	0.1	5.1	−4.5	−5.8	3.3	−0.2	−6.3
O_2_-concentration error by the direct method (%)	23.3	−42.4	20.9	10.2	−65.0	15.7	−42.3
O_2_-concentration error by the modified method (%)	13.6	6.1	15.9	−0.2	−11.9	1.9	−11.7

## Data Availability

Not applicable.

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
