# Peer review of "Resolving Cross-Sensitivity Effect in Fluorescence Quenching for Simultaneously Sensing Oxygen and Ammonia Concentrations by an Optical Dual Gas Sensor"

_sensors, 2021, doi:10.3390/s21206940_

Round 1

Reviewer 1 Report

The manuscript is dedicated to fabrication and testing of model two-probe sensor for quantitative determination of O2 and NH3 gases. The main goal of this work is demonstration of applicability of such method for accurate estimation of gas concentration. The author fabricated paper-based device with Eosin Y and PtTFPP probes for NH3 and O2, respectively. Effects of gas concentration was studied for pure gases and the mixtures with various ratio. Based on their studies, authors suggested a method for measurements of gas concentration with relatively high accuracies.

The text of manuscript is well-written and thoroughly describes experimental details and discussion of obtained results.

However, two comments were arisen after reading the text.

1) The abstract is too general. Author should provide more specific information such as sensor fabrication, used substances in abstract.

2) Author should indicate the final concentrations of fluorescent probes in devices. Have authors screened the concentration of probes and the ration between Eosin Y and PtTFPP on sensing properties? Author should comment this moment.

The manuscript could be published after this minor revision.

Reviewer 2 Report

Liu et al. reported an analysis assay used for dual of oxygen and ammonia determination, which can decrease the cross-sensitivity effect in the multi-analytes detection process.

  1. Line 43–45: “In addition, its detection sensitivity is higher than that of other spectroscopic methods based on optical measurements of absorption, reflection, interference, Raman scattering, and surface plasmon resonance” The sentence is lack enough evidence.
  2. Authors should describe Section 3.2 firstly and then is 3.1.
  3. Line 201–202: “Figure 5 shows an emission spectrum (black curve) under 40% of O2 and 400 ppm of NH3” Why did the authors choose the conditions?
  4. Section 3.4, 3.6: the range scope and LOD for single gas and mixture gas monitoring should be supplemented and compared.
  5. In my opinion, Section 3 is difficult to read, and thus losing readers' attention. In addition, it would be nice to use more concise language with academic style.

6.Figure 8 and 9 can be combined in one figure.

Round 2

Reviewer 2 Report

All points are fully addressed and revised by the authors, thereby no further corrections needed.